# PKCβ Facilitates Leukemogenesis in Chronic Lymphocytic Leukaemia by Promoting Constitutive BCR-Mediated Signalling

**DOI:** 10.3390/cancers14236006

**Published:** 2022-12-06

**Authors:** Jodie Hay, Anuradha Tarafdar, Ailsa K. Holroyd, Hothri A. Moka, Karen M. Dunn, Alzahra Alshayeb, Bryony H. Lloyd, Jennifer Cassels, Natasha Malik, Ashfia F. Khan, IengFong Sou, Jamie Lees, Hassan N. B. Almuhanna, Nagesh Kalakonda, Joseph R. Slupsky, Alison M. Michie

**Affiliations:** 1School of Cancer Sciences, College of Medicine, Veterinary and Life Sciences, University of Glasgow, Glasgow G12 8QQ, UK; 2Paul O’Gorman Leukaemia Research Centre, Gartnavel General Hospital, 21 Shelley Road, Glasgow G12 0ZD, UK; 3Institute of Systems, Molecular and Integrative Biology, University of Liverpool, Liverpool L69 7BE, UK

**Keywords:** CLL, PKCβ, SP1, BCR signalling, leukemogenesis

## Abstract

**Simple Summary:**

Chronic lymphocytic leukaemia (CLL) is the most common blood cancer in the Western world and remains incurable. While a cause for this cancer has not been defined, one specific protein, protein kinase CβII (PKCβII), is highly expressed in CLL cells and is linked with a poorer clinical outcome. This study shows that PKCβ expression plays a central role in assisting the development of leukemic cells in our CLL mouse model. Moreover, a ubiquitously expressed protein SP1 is important for driving the genetic program that promotes leukaemia in our model system, and this program is assisted by PKCβ. Importantly, this SP1-driven program is also observed in human CLL cells, suggesting a role for PKCβ in the development of human disease.

**Abstract:**

B cell antigen receptor (BCR) signalling competence is critical for the pathogenesis of chronic lymphocytic leukaemia (CLL). Defining key proteins that facilitate these networks aid in the identification of targets for therapeutic exploitation. We previously demonstrated that reduced PKCα function in mouse hematopoietic stem/progenitor cells (HPSCs) resulted in PKCβII upregulation and generation of a poor-prognostic CLL-like disease. Here, *prkcb* knockdown in HSPCs leads to reduced survival of PKCα-KR-expressing CLL-like cells, concurrent with reduced expression of the leukemic markers CD5 and CD23. SP1 promotes elevated expression of *prkcb* in PKCα-KR expressing cells enabling leukemogenesis. Global gene analysis revealed an upregulation of genes associated with B cell activation in PKCα-KR expressing cells, coincident with upregulation of PKCβII: supported by activation of key signalling hubs proximal to the BCR and elevated proliferation. Ibrutinib (BTK inhibitor) or enzastaurin (PKCβII inhibitor) treatment of PKCα-KR expressing cells and primary CLL cells showed similar patterns of Akt/mTOR pathway inhibition, supporting the role for PKCβII in maintaining proliferative signals in our CLL mouse model. Ibrutinib or enzastaurin treatment also reduced PKCα-KR-CLL cell migration towards CXCL12. Overall, we demonstrate that PKCβ expression facilitates leukemogenesis and identify that BCR-mediated signalling is a key driver of CLL development in the PKCα-KR model.

## 1. Introduction

Intracellular signals transmitted through the B cell antigen receptor (BCR) regulate B cell differentiation, survival and proliferation at distinct stages of development, maturation and activation, and play a critical role in driving the pathogenesis of chronic lymphocytic leukaemia (CLL) [1]. The BCR is a critical prognostic marker of CLL, with sequences aligning closely with germline (>98% similarity; unmutated IgHV) generally associated with poor prognostic CLL cases [2]. The structure of the BCR appears to be related to its ability to transmit and activate intracellular signalling networks more efficiently, thus promoting CLL cell survival and proliferation [3]. BCR crosslinking promotes membrane localisation and activation of a number of cytoplasmic protein kinases including Lyn and spleen tyrosine kinase (SYK), and activates Bruton’s tyrosine kinase (BTK), triggering the activation of multiple key signalling pathways and transcription factors that regulate B cell proliferation, differentiation and apoptosis in a cell maturation-dependent manner [4].

Membrane-recruited BTK undergoes auto-phosphorylation and then activates phospholipase Cγ2 (PLCγ2) [5] which further catalyses the hydrolysis of phosphatidylinositol 4,5-bisphosphate (PIP_2_) into diacylglycerol (DAG) and inositol trisphosphate (IP_3_). Intracellular calcium levels increase as a result of IP_3_ production, thus providing the cofactors required for classical protein kinase C (PKC: PKCα, βI/II and γ) isoform activation [6]. PKC isoforms, specifically PKCβII in B cells, activate survival pathways through NF-κB-mediated signalling [7], thus linking proximal BCR-mediated signals with downstream pathways. CLL cells exhibit a dysregulated PKC isoform expression profile: upregulation of PKCβII, PKCε, PKCζ and downregulation of PKCα and PKCβI compared to normal B cells, with the increased levels of PKCβII expression correlating with the poor prognostic outcome for CLL patients [8]. Furthermore, ZAP-70 recruits PKCβII to lipid rafts in CLL cells, which activates and enhances the translocation of PKCβII to the mitochondria where it phosphorylates the anti-apoptotic protein BCL2 [9].

PKCβ has been shown to play an essential role in the regulation of leukaemia development in the TCL1 mouse model of CLL, as indicated by the abrogation of leukemic cells in PKCβ KO mice [10]. We developed a CLL-like mouse model, by introducing a kinase-inactive PKCα (PKCα-KR) construct in mouse hematopoietic stem/progenitor cells (HSPCs), which resulted in the development of an aggressive subset of CLL in vitro and in vivo, exhibiting an upregulation of ZAP-70, enhanced proliferation and increased tumour load in the lymphoid organs [11]. Analysis of the PKC isoform profile in PKCα-KR-transformed cells revealed that similar to that seen in primary CLL samples, PKCβII expression was upregulated, and this occurred at later stages of disease development [12]. Here, we determine the importance of PKCβ expression in disease development in our mouse model, analyse the mechanisms that promote PKCβ upregulation and delineate the signalling pathways that occur downstream of BTK/PKCβ, which may have particular importance in developing future therapies for B cell malignancies.

## 2. Materials and Methods

### 2.1. Primary Cells and Cell Lines

HSPCs obtained from the foetal liver (FL) of wild-type ICR mice on day 14 of gestation were prepared as described previously, and retrovirally transduced with a plasmid encoding either empty vector control (MIEV) or MIEV-PKCα-KR (PKCα-KR) with the bicistronic expression of GFP. Maintenance of the retroviral constructs within the cells during the cultures was assessed by GFP expression using flow cytometry [11]. Immediately prior to analysis, MIEV and PKCα-KR cells were removed from OP9 monolayers and placed on empty 6 well plates for 2 h to remove carry-over OP9 cells. All cell lines were routinely tested for mycoplasma contamination. All mice were maintained at the University of Glasgow Central Research Facilities under standard animal housing conditions in accordance with local and home office regulations. Primary CLL cells, obtained from patients that had given informed consent, were isolated as described previously [13] and cryopreserved for future use. Clinical details of patients used in these studies are presented in Appendix A; none of the patients had received chemotherapy within the preceding 3 months.

### 2.2. Surface and Intracellular Staining and Flow Cytometric Analysis

Flow cytometry reagents used are detailed in Appendix A, including the mouse CLL surface markers (CD19-APC-Cy7, CD5-APC, CD23-PE-Cy7, CD45-PerCP) and phospho-BTK antibodies (BTK^Y223^-AF647, BTK^Y551^-AF647). After drug treatment/stimulation as indicated, MIEV and PKCα-KR cells (≥1 × 10^6^ cell/condition) were stained as described previously [11]. Cells were acquired using a FACSCantoII flow cytometer with BD FACSDiva software and analysed using FlowJo (Tree Star Inc., Ashland, OR, USA) software.

### 2.3. Cell Stimulation/Drug Treatment

MIEV and PKCα-KR cells (≥1 × 10^6^/condition) were treated with 1 µM ibrutinib (IB) or 20 µM enzastaurin (Enza) or vehicle control in 10% FBS/αMEM media as previously published [12]. Primary CLL cells were pre-incubated with 1 µM IB, 10 µM Enza or vehicle control in 10% FBS/RPMI media on ice and then stimulated with 10 µg/mL F(ab’)_2_ fragment anti-human IgM (Stratech Scientific Ltd., Ely, UK) to crosslink the BCR (BCR-XL) at 37 °C and then analysed.

### 2.4. Cell Counting Beads

To compare cell counts between different co-culture conditions, a set volume of CountBright beads (BD Biosciences, Wokingham, UK) was added to each PKCα-KR sample as indicated, prior to flow cytometry. After FSC/SSC gating on the beads, 2000 beads were acquired together with a variable cell number to enable a relative cell number to be calculated between samples [14].

### 2.5. Proliferation/Apoptosis Assays

MIEV and PKCα-KR cells (≥1 × 10^6^/condition) were labelled with CellTrace Violet (CTV; 2 µM) using CellTrace™ Cell Proliferation Kits (Life Technologies, Paisley, UK) as described previously [14]. Cells were analysed on the FACSCantoII flow cytometer. Results are expressed relative to the mean fluorescence intensity (MFI) of CTV in no drug control (NDC) cells at 48 h. At the end of the stimulations/drug treatment incubations, cells were stained with Annexin V/7-AAD as described previously [13]. 

### 2.6. Lentiviral Knockdown of prkcb

Lentiviral knockdown vectors targeting mouse *prkcb* and non-targeting control (SCR) were purchased from Sigma-Aldrich (St. Louis, MO, USA) in the pLKO.1 backbone. Lentiviral particles were generated using a three-plasmid HEK293T lentiviral protocol, as described previously [15]. Briefly, HEK293 cells were transfected overnight using CaCl_2_ and 2 × HBS containing pHIV1 and VSV-G as accessory plasmids. The virus was collected in DMEM supplemented with 20% FCS and freshly isolated HSPCs (d0) were cultured in a viral medium supplemented with 4 μg/mL polybrene and IL7/Flt3 (10 ng/mL each). This was repeated 4 times over 48 h. After transduction, the cells were washed and re-suspended in complete media. Cell sorting for GFP^+^CD19^+^CD45^+^CD11b^−^ cells was performed using FACSAria^TM^ III cell sorter (BD Biosciences) on day 7; MIEV and PKCα-KR were then introduced by retroviral transduction on day 8 as described previously [11].

### 2.7. qPCR

RNA was extracted from MIEV- or PKCα-KR-transduced cells from day 17–23 co-cultures using RNeasy mini kit (Qiagen, Manchester, UK) according to the manufacturer’s protocol. Synthesis of cDNA was performed using SuperScript^®^ III Reverse transcriptase (Life Technologies) as per the manufacturer’s protocol. Real-time PCR (qPCR) was conducted using the TaqMan PCR Master Mix (Applied Biosystems, Warrington, UK), with glyceraldehyde-3-phosphate dehydrogenase (*gapdh*) used as a reference control. Inventoried primers and probes and PCR buffers were purchased from Applied Biosystems unless otherwise stated. *Egr1* and *Btk* primers were designed and optimised, using TATA-box binding protein (*tbp*) as a reference control (Appendix A). For each PCR reaction, 1 µL cDNA was used followed by an appropriate primer master mix containing 0.25 µM forward, 0.25 µM reverse primers and 2xSYBR Green PCR Mastermix (ThermoFisher Scientific, Paisley, UK). Technical triplicates of each PCR reaction were performed. qPCR was performed on the 7900HT Fast Real-Time PCR System (Applied Biosystems): The cycle condition for 40 cycles is as follows: 95 °C 2 min, 40 cycles of 95 °C 5 s, 60 °C 10 s, 72 °C 5–20 s. All primer pairs were optimised to ensure efficient amplification of a single PCR product. NDC or MIEV was used as the calibrator and data were analysed using the 2^−ΔΔCT^ relative quantification method.

### 2.8. Chromatin Immunoprecipitation (ChIP)

MIEV- and PKCα-KR-transduced cells were co-cultured on OP9 cells until days 20–23, as described above. The cells were harvested and IP was performed on the sonicated chromatin material using a ChIP grade antibody—anti-SP1 antibody (Clone D4C3; Cell Signaling Technologies, Danvers, MA, USA) or IgG (negative control), as described previously [16]. The primer sets were designed on regions flanking three SP1 binding regions within the *prkcb* promoter (Appendix A).

### 2.9. Microarrays

MIEV- or PKCα-KR-transduced HSPCs were co-cultured on OP9 cells in the presence of IL-7 for 17–23 days (late co-culture) and total RNA was isolated using an RNeasy kit (Qiagen, Manchester, UK) from five independent co-cultures. The RNA was quantified using a Nanodrop ND-1000 Spectrophotometer (ThermoFisher Scientific). Screening of the RNA samples was performed against Affymetrix^TM^ GeneChip^®^ Mouse Gene 1.0 ST Array (Santa Clara, CA, USA). Background correction and normalisation of generated CEL files were performed using Robust Multichip Average (RMA) in Partek Genomics Suite 7.19.1125 (St. Louis, MO, USA). Differentially expressed genes, PKCα-KR vs. MIEV, were detected by one-way analysis of variance. Significant genes (*n* = 2836), were identified with a fold change ±1.2 and *p*-value < 0.05, with 1513 up- and 1323 downregulated in PKCα-KR vs. MIEV.

### 2.10. Western Blots

Cell lysates were prepared in lysis buffer (1% Triton, 1 mM 1,4-dithiothreitol (DTT), 2 mM ethylenediaminetetraacetic acid (EDTA), 20 mM Tris pH 7.5 containing a complete protease inhibitor, and PhosStop; Roche) from early (day 6–12) or late (day 15–23) MIEV and PKCα-KR cells, or primary CLL cells, with or without drug treatment as indicated and quantified using the Bradford Assay (ThermoFisher Scientific). The antibodies used are described in Appendix A, and Western blotting was performed as described previously [14]. Images were developed using an SRX 101A film processor.

### 2.11. Migration Assay

MIEV- or PKCα-KR-transduced cells (2 × 10^5^) were incubated in 100 μL RPMI-1640/0.5% BSA media in the presence or absence of drugs (as indicated) for 30 min prior to the assay. Cells were then transferred to the upper chamber of a 6.5 mm diameter Transwell culture insert (Costar^®^, Fisher Scientific, Loughborough, UK.) and placed into wells containing 600 μL media supplemented with 150 ng/mL CXCL12. The cells were incubated for 4 h at 37 °C. Thereafter, three 150 μL aliquots were removed from each lower chamber for counting by flow cytometry. The total number of events acquired during 20 s on high flow setting was recorded for each aliquot.

### 2.12. Statistics

*p* values were determined by students paired or unpaired *t*-test or mixed model ANOVA on a minimum of at least 3 biological replicates to compare data using GraphPad Prism 6 software (GraphPad Software Inc., San Diego, CA, USA) as indicated; *, **, *** and **** represent *p* < 0.05, 0.01, 0.001 and <0.0001, respectively. Biological replicates were derived from individual cell culture conditions from distinct biological samples. Results are presented as mean ± standard error of the mean of biological replicates (SEM), with the number of replicates performed highlighted in the figure legends.

## 3. Results

### 3.1. PKCβ Promotes Leukemogenesis in the PKCα-KR CLL-like Mouse Model

We and others have previously demonstrated that PKCβII is upregulated in CLL mouse models (Eμ-Tcl-1 and PKCα-KR; [12]) and in primary CLL samples, particularly in poor prognostic samples suggesting a role in CLL pathogenesis [8]. To extend this research we addressed the importance of *prkcb* expression during the initiation of the CLL-like disease in our PKCα-KR mouse model. Performing *prkcb* shRNA knockdown (KD) experiments prior to retroviral transduction with PKCα-KR revealed that the generation of the leukemic phenotype was negatively impacted upon the reduction in *prkcb* expression. There was a significant reduction in surface expression of the leukemic marker characteristic of this CLL model, CD23 and CD5 (Figure 1A,B, Appendix A), as well as CD45 expression. An additional feature of PKCα-KR-transduced cells is an increased cellular count in the cultures, which was significantly reduced, together with elevated apoptosis in *prkcb* KD conditions (Figure 1C,D). A significant reduction in PKCβ mRNA levels was confirmed in cultured cells transduced with sh-*prkcb*, compared with SCR controls (Figure 1E). The reduced cell numbers generated with PKCα-KR-transduced *prkcb* KD cells precluded us from testing whether CLL-like development was blocked in vivo. Taken together, these results indicate that PKCβ plays an important role in the initiation of CLL in our mouse model, supporting previous findings in the Eμ-Tcl-1 CLL mouse model [10].

### 3.2. Sp1 Regulates Similar Transcriptional Networks in the PKCα-KR CLL-like Cells and Primary Human CLL Samples

The transcription factor SP1 plays a central role in *PRKCB* gene regulation in human CLL [17]. We were interested in determining whether the *prkcb* gene was similarly regulated by Sp1 in our CLL mouse model. The *prkcb* promoter contains three Sp1 binding regions (Figure 2A). ChIP analysis revealed that Sp1 binding to the *prkcb* promoter was significantly upregulated in PKCα-KR transduced CLL-like cells at all three Sp1 binding sites, compared with MIEV cells (Figure 2B). These data were supported with experiments using mithramycin (mtm), an anti-cancer antibiotic that selectively binds G-C-rich DNA sequences to inhibit RNA and DNA polymerases and globally displace Sp1 [18]. Treatment of PKCα-KR cells with 200 nM mtm significantly reduced Sp1 binding activity at all regions assessed on the *prkcb* promoter (Figure 2C). We have previously shown that PKCβII is upregulated in PKCα-KR cells compared with MIEV cells [12], and mtm treatment resulted in a downregulation of PKCβII protein expression specifically in PKCα-KR cells, although this downregulation did not reach significance (Figure 2D, Appendix A). Assessing the wider impact of mtm treatment on transcriptional changes in our PKCα-KR cells revealed a reduction in expression not only of *prkcb*, but also *Sp1* itself and *Bcl2*, *Vegfa*, *Blnk* and *Lef1* upon treatment with mtm, compared to no drug control (NDC; Figure 2E), a result similar to that seen in human CLL samples (Appendix A). Taken together, these data suggest a similar regulation of Sp1-mediated transcription networks between poor prognostic human CLL cells and PKCα-KR CLL cells.

### 3.3. BCR Signalling Components Are Dysregulated in PKCα-KR CLL-like Cells

To investigate global gene expression profiles in the PKCα-KR CLL mouse model within the timeline of PKCβII upregulation, microarrays were performed comparing MIEV- and PKCα-KR-transduced cells at the late stage of B cell transformation (post d17 of co-culture after retroviral transduction with MIEV- and PKCα-KR-constructs at d0 [12]). Analysis of the dataset revealed differentially regulated gene expression between the MIEV- and PKCα-KR-transduced cells, and enrichment of pathways that were significantly dysregulated, with immune system processes identified as the most enriched group of pathways (Figure 3, Appendix A). This analysis indicated an upregulation of genes allied to B cell activation in PKCα-KR-transduced cells, and given the importance of BCR signalling in the pathology of CLL, we chose to validate this pathway in our CLL mouse model (Figure 4). Deregulation of the BCR signalling in PKCα-KR-transduced cells, summarised in Figure 4A, is highlighted by a number of features including CD45 and CD19 upregulation in the CLL-like PKCα-KR cells (indicated in pink; previously confirmed by flow cytometry [12]), together with an upregulation of signalling components proximal to BCR-mediated signalling (*Lyn*, *Btk*, *Blnk*) while *Syk*, *Pten* and *Pdk1* are downregulated (indicated in green) compared to MIEV control cells. The expression of some transcription factors was dysregulated, with PKCα-KR cells showing upregulation of *Bcl6*, *Egr1*, *Elk1*, *NFkB* and *Foxo1*, while *Creb* and *Tcf1* were downregulated compared to MIEV control cells in the microarray datasets. Upregulation of *Egr1* and *Btk* in the late PKCα-KR cultures was confirmed by qPCR (Figure 4B). In support of the data shown in Figure 2E, *Prkcb* and *Lef1* expression patterns aligned with the *Sp1* expression, with downregulation of these genes in early PKCα-KR cultures, and significant upregulation in the late PKCα-KR cultures, compared to the respective MIEV cells (Figure 4C).

Analysing the expression levels of BCR signalling components demonstrated that key hubs proximal and distal to the BCR were dysregulated in the CLL-like PKCα-KR cells compared to MIEV control cells both at the early and late stages of the CLL-like disease. Lyn, Egr1 and c-Myc expressions were upregulated, while Lck was downregulated (Figure 5A,C, Appendix A). PKCα expression was significantly downregulated in the PKCα-KR cells, in agreement with our previous work [12]. Furthermore, a significant elevation in pAKT^S473^ and pS6^S235/236^ was observed, indicating an elevation in AKT/mTOR-mediated signalling in PKCα-KR cultures (Figure 5A, Appendix A). An upregulation in pc-Myc^S62^ was also observed, accompanied by an increase in c-Myc expression in PKCα-KR cells compared with MIEV suggesting a stabilisation of c-Myc in PKCα-KR cells; however, these modulations did not reach significance (Figure 5C, Appendix A). Phospho-flow staining revealed a significant elevation of both BTK^Y551^ and BTK^Y223^ phosphorylation in PKCα-KR compared to MIEV cells (Figure 5B). Taken together these results indicate that PKCα-KR CLL-like cells exhibit upregulated BCR signalling activity.

### 3.4. PKCα-KR CLL-like and Primary Human CLL Cells Share Similar Responses to BCR-Targeted Inhibitors

To determine how PKCβ impacted BCR signalling, MIEV- and PKCα-KR-expressing cells were treated with either BTK inhibitor ibrutinib (IB) or PKCβII inhibitor enzastaurin (Enza). While the drug treatments did not affect the BTK phosphorylation on MIEV B-cells, IB- and Enza-treatment significantly reduced BTK^Y223^ phosphorylation suggesting an inhibition in BTK activity in the PKCα-KR-expressing B cells (Figure 5B). The elevation in S6^S235/236^ phosphorylation in PKCα-KR-expressing cells was inhibited by Enza treatment. The elevation in c-Myc^S62^ phosphorylation and c-Myc expression in PKCα-KR cells was not significantly affected by drug treatments (Figure 5C, Appendix A). These findings suggest that PKCβ plays a vital role in driving the activation of key proteins within the BCR-mediated BTK and AKT/mTOR signalling in PKCα-KR cells.

To establish whether our findings in the PKCα-KR CLL-like model were applicable to primary human CLL samples, we treated human CLL cells with IB or Enza in the presence of F(ab’)_2_ stimulation (BCR-XL) and assessed the phosphorylation/activation status of BTK. Both BTK^Y551^ and BTK^Y223^ phosphorylation levels were significantly downregulated upon PKCβ inhibition with Enza in human CLL cells compared to NDC, while IB treatment led to a significant downregulation of BTK^Y223^ phosphorylation only (Figure 6A). Western blotting of primary CLL cell lysates stimulated with BCR-XL showed an elevation in phosphorylation of AKT^S473^ and pS6^S3235/236^ and an increase in c-Myc expression similar to that noted in PKCα-KR cells. Furthermore, a reduction in AKT^S473^ and pS6^S3235/236^ phosphorylation was more pronounced with Enza treatment in the presence of BCR-XL, although this did not reach significance. Enza significantly decreased overall c-Myc^S62^ phosphorylation upon BCR-XL; however, coupled with the significant reduction in total c-Myc protein with IB or Enza treatment, this led to a significant increase in pc-Myc^S62^ phosphorylation on the remaining c-Myc in human CLL cells (Figure 6B, Appendix A), findings that mirrored trends in PKCα-KR CLL-like cells.

We have previously shown that Enza significantly and selectively impacts PKCα-KR cell viability (20 μM) and proliferation (10 μM) over MIEV cells [12]. Testing cell viability with increasing IB concentrations revealed that MIEV cells were more sensitive to IB than PKCα-KR cells up to a concentration of 10 μM IB (Appendix A). MIEV and PKCα-KR cells were similarly affected by IB treatment with a significant block in proliferation at 3 μM (Figure 6C). As it has previously been established that inhibition of BCR-mediated signals with IB can reduce CLL cell migration towards CXCL12 [19], and leukocyte migration was the fourth most dysregulated pathway in PKCα-KR cells, the migration capacity of MIEV and PKCα-KR CLL-like cells was tested in the presence of IB or Enza treatment. Initial analysis of the adhesion marker CD38, which is regulated by BCR-mediated signalling, showed that surface expression of CD38 was upregulated on PKCα-KR cells compared with MIEV cells (Figure 6D) [20,21]. Our data revealed that PKCα-KR cells have a significantly enhanced migration ability towards SDF1 compared to MIEV cells (Figure 6E), likely due to elevated surface expression of CXCR4 and CD38 (Figure 6D, Appendix A), and elevated BCR-mediated signalling. However, while migration was significantly inhibited upon treatment with either IB or Enza, these treatments did not alter the surface expression of CD38 (Appendix A).

## 4. Discussion

BCR signals are central to CLL pathogenesis, and PKCβ is an important regulator of these signals in healthy and malignant B cells. Although we have known for many years that PKCβII is overexpressed in the malignant cells of CLL [8], the link between PKCβ and leukemogenesis is not yet demonstrated. Here, we show that PKCβ expression plays a central role in the development of leukemic cells in our model of CLL (PKCα-KR mouse model). Moreover, we show that the transcription factor SP1 is important for driving a transcription program that promotes leukemogenesis in our model system, and that this program is, at least in part, driven by PKCβ activity. Importantly, this SP1-driven program is also observed in human CLL cells, suggesting a role for PKCβ in the pathogenesis of the human disease.

The critical role played by PKCβ in the development of normal B cells has long been established [22]: targeted disruption of the *prkcb* gene in mice results in a severe reduction in marginal zone and B1 populations, while our own work shows that overexpression of PKCβII expands these B cell populations [23]. In contrast, the PKCα knockout mouse model has no phenotype in early B lymphocyte development or proliferation and to date, this model has not been associated with the development of haematological malignancies [24]. Indeed, PKCα-deficient mice display an increased risk of developing colorectal cancer which similar to CLL, exhibits a reduction in PKCα expression and an elevation in PKCβII expression in patient samples [25]. In the current study, we demonstrate that both CD5 and CD23 were downregulated in PKCα-KR-transduced cells on a background of *prkcb* KD in vitro. This is distinct from the PKCβ-deficient mice, which only exhibited a reduction in CD5^+^ B1 B cells, with surface CD23 expression on B cell populations being unaffected [22]. This indicates that *prkcb* KD targets the leukemic phenotype present in the PKCα-KR mouse model. A relationship between PKCβ and CLL pathogenesis was suggested in studies showing overexpression of this isoform in the malignant cells of this disease, particularly in cells from patients with late-stage disease [8]. More recently, PKCβ was found to play a critical role in disease development in the Eμ-TCL1 CLL mouse model, with no leukaemia development in the absence of PKCβ expression [10]. Subsequent studies of this model highlighted the essential role played by PKCβII within the stromal compartment in promoting both maintenance and chemosensitivity of malignant B cells [26,27] but did not address whether this PKC isoform was involved in the initiation of malignant cell transformation. Our work provides clarity to this question and shows that PKCβ expression is important for leukemogenesis in our model of CLL, a model we have previously shown is similar to human CLL with respect to overexpression of PKCβII in the leukemic cells [12]. It is likely, however, that the role of PKCβ in this context is one of facilitation rather than transformation because our earlier work has also shown malignant disease does not occur when this PKC isoform is specifically overexpressed in B cells [27]. A similar role for PKCβ overexpression has been suggested for neoplastic transformation in colon cancer [25,28]. Another BCR pathway protein overexpressed in CLL cells, Lyn, is similar to PKCβ in promoting the expansion of the malignant clone through its role within microenvironmental cells; however, it differs because Lyn deficiency does not affect the rate of malignant cell transformation in the Eμ-Tcl1 murine model of CLL [29]. Thus, the current study remains the first to directly implicate a signalling protein within the BCR pathway in CLL leukemogenesis.

Our data show that elevated PKCβ expression in PKCα-KR-transduced cells is driven by elevated Sp1 binding activity at the *prkcb* promoter. This is similar to what is observed in human CLL cells [17] and suggests that the mechanisms controlling PKCβ expression are accurately modelled by our system. A further similarity is demonstrated by our experiments using mtm to target the DNA binding activity of SP1. In addition to *prkcb* downregulation, mtm treatment reduces expression of *blnk*, *bcl2*, *vegfa*, *lef1* and *sp1* in the PKCα-KR-transduced cells, and mtm treatment of CLL cells results in comparable downregulation of the human orthologues of these genes. Sp1, a ubiquitous transcription factor, may play a significant role in promoting a gene transcription profile that induces CLL cell survival and is aligned with poor prognostic disease [30]. This notion is supported by recent work showing that targeted delivery of miR-29b to CLL cells leads to enhanced cellular reprogramming and decreased viability due to its effect of decreasing SP1 expression [31]. Furthermore, mtm analogues have been shown to be effective against CLL cells, with EC-7072 inducing CLL cell death through targeting of BCR signalling [32], and administration of MTM_OX_32E to the Eμ-Tcl1 mouse model reduces the malignant cell burden [33].

BCR signalling strength in malignant cells of CLL correlates with poor disease prognosis. This has relevance to our model because it was one of the top dysregulated pathways in our microarray comparing gene expression between PKCα-KR-transduced and control cells. Biochemical analysis of the activation status of signalling pathways downstream of the BCR indicated that our mouse model possesses constitutively active BCR signalling, as shown by elevated phosphorylation of BTK^Y223^, AKT^S473^, S6^S235/236^ and c-Myc^S62^, and increased expression of the early activation marker EGR1. We further noted a downregulation in Lck expression which has previously been described in BCR-activated CLL cells [34]. These findings validate our microarray data and consolidate our previous findings showing activation of the ERK-MAPK- and AKT/mTOR signalling pathways in PKCα-KR-expressing cells [11,12,35]. The significant upregulation of both BTK^Y551^ and BTK^Y223^ phosphorylation in our mouse model indicates that BTK is constitutively active, leading to activation of distal BCR-mediated signalling (i.e., ERK-MAPK pathway), and suggests that such phosphorylation may be aided by elevated Lyn kinase expression which is reported to target phosphorylation of BTK^Y551^ and promote downstream BCR signalling events [36]. A role for PKCβ in regulating this phosphorylation event is shown in experiments using enzastaurin or ibrutinib. Both compounds inhibited BTK^Y223^ phosphorylation in PKCα-KR and primary CLL cells, causing a reduction in AKT/mTOR and ERK-MAPK/c-Myc mediated pathways. One difference between PKCα-KR and primary CLL cells in this respect is that enzastaurin also inhibited BTK^Y551^ phosphorylation in the latter but not the former. While this result may be related to the off-target effects of enzastaurin [37], it may suggest that PKCβII may have an impact on BCR signalling upstream of BTK activation. While the mechanism regulating PKCβII-mediated phosphorylation of BTK is not clear, it is interesting to note that PKCβII has been demonstrated to negatively regulate BTK by S180 phosphorylation to prevent its membrane recruitment [38]. Whether these two events are linked remains to be elucidated in future studies.

While it has been well established that ibrutinib has enhanced the survival of patients with B cell malignancies including CLL, mantle cell lymphoma and Waldenström’s macroglobulinemia (WM) [39,40,41], clinical trials incorporating enzastaurin have not, as yet, revealed any significant improvement in patient survival in DLBCL [42]. Pre-clinical studies using sotrastaurin, a more broad-range PKC isoform inhibitor targeting α, β and θ isoforms, in CLL and DLBCL indicate a strong anti-tumour effect [43,44]. While the reasons for the lack of effect of enzastaurin are not entirely clear, it may be due to the study design as it was used as relapse prevention rather than as disease-active therapy in high-risk patients. Given the interest in developing appropriate models to test potential therapeutic compounds, we consider that our CLL-like mouse model provides an excellent model for carrying out pre-clinical testing of novel BCR-targeted agents, that may be translated towards the clinic.

## 5. Conclusions

The data reported here demonstrate a role for PKCβ in the development of leukemic B cells in our model and suggest it may be important for facilitating leukemogenesis in CLL. This has implications for the current understanding of CLL pathogenesis where signalling through the BCR pathway is important. Although many of the kinases in this pathway are overexpressed in CLL cells, PKCβ is the only one currently that facilitates the neoplastic transformation of these cells. Moreover, the current study validates our PKCα-KR model for the study of CLL cells, where many responses observed using PKCα-KR CLL cells are similar to that of primary CLL cells. Given this parallel, we consider that this CLL-like mouse model will be an excellent tool to decipher the pathobiological behaviour of CLL cells.

## Figures and Tables

**Figure 1 cancers-14-06006-f001:**
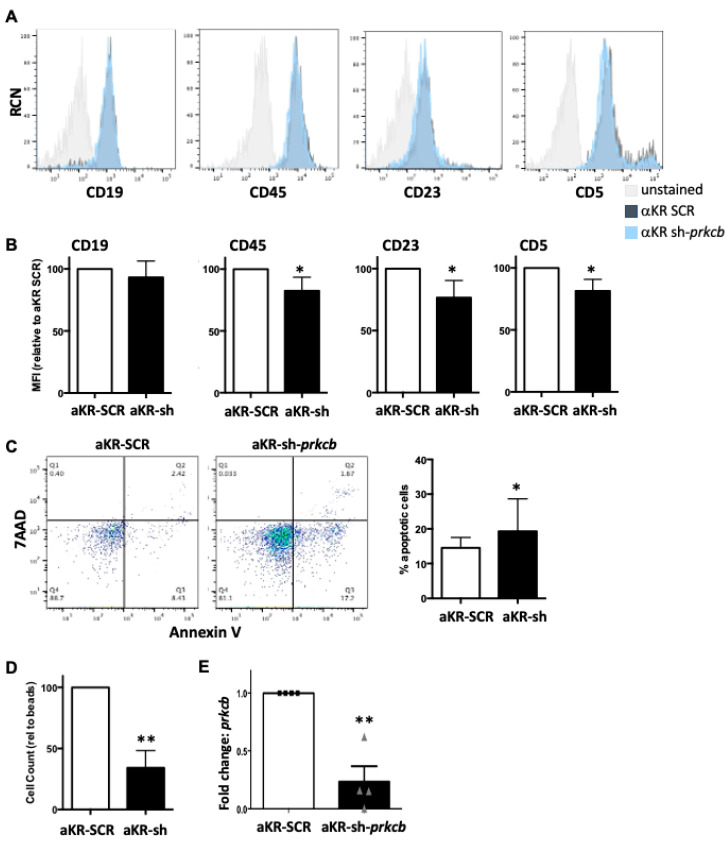
Reduction in PKCβ expression inhibits the initiation of PKCα-KR-mediated CLL development. Knockdown of *prkcb* (or scrambled (SCR) control) was performed in HSPC cells within 24 h of isolation from the mouse, then these cells were retrovirally transduced with MIEV or PKCα-KR (αKR) at d7. Cells were co-cultured with OP9 in the presence of cytokines for up to 35 days. Phenotypic characterisation of the cells was carried out by flow cytometry analysing: (**A**) CD19, CD45, CD23 and CD5. Representative histogram plots are shown, gated on FSC/SSC and GFP+ cells comparing αKR sh-*prkcb* cells with αKR-SCR cells compared with unstained cells (gating strategy shown in Appendix A); (**B**) Average MFIs of CD19, CD45, CD23 and CD5 surface markers are shown relative to αKR-SCR cultures (*n* = 5); (**C**) Apoptosis was determined by AnnV/7AAD staining. A representative dot plot shows Annexin V vs. 7AAD staining in αKR-SCR cells and αKR sh-*prkcb*. The graph shows the percentage apoptotic (AnnV+) cells present in cultures post d14 (*n* = 5); (**D**) Cell counts were performed with flow cytometry using counting beads, shown relative to a set bead number acquired (*n* = 5); (**E**) qPCR analysis of *prkcb* expression, αKR-SCR (circles) compared with αKR sh-*prkcb* (triangles). *Gapdh* was used as the reference gene and αKR-SCR-transduced cells were used as a calibrator (*n* = 4). All experiments shown are representative of *n* ≥ 4 biological replicates as indicated. Paired student *t*-test with Wilcoxon matched-pair signed rank test was used to analyse the data. * *p* < 0.05, ** *p* < 0.01.

**Figure 2 cancers-14-06006-f002:**
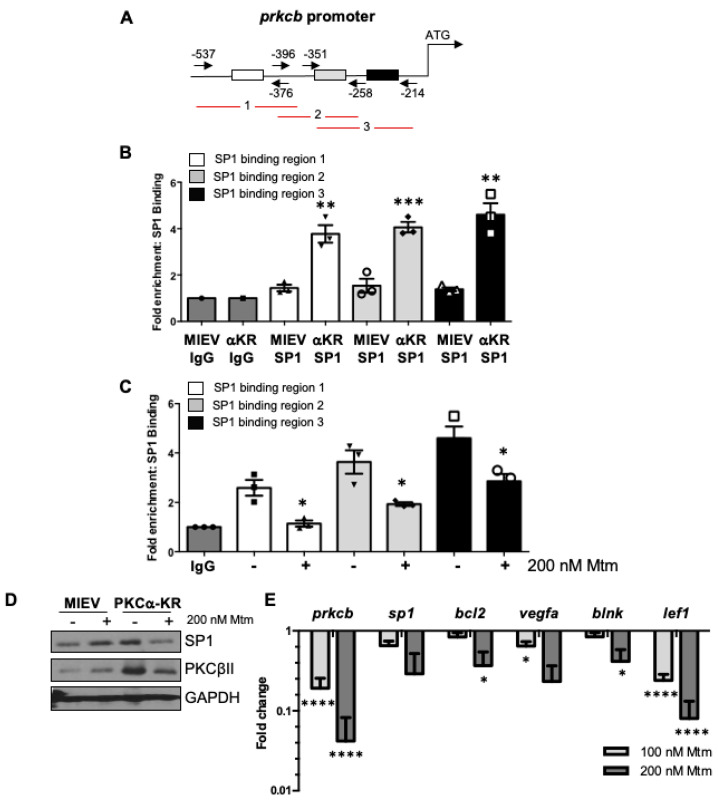
Sp1 binding is upregulated in the *prkcb* promoter of PKCα-KR transduced CLL-like cells. (**A**) Diagram showing Sp1 binding sites at the *prkcb* promoter, the primer locations and the three products; (**B**) ChIP analysis was performed in MIEV (binding region 1 (black triangles), 2 (white circles), 3 (white triangles)) and PKCα-KR cells (binding region 1 (inverted triangles), 2 (diamonds), 3 (squares)) at the late stage of co-culture (> d17) to determine Sp1 binding occupancy at the *prkcb* promoter (average of *n* = 3 biological replicates); (**C**) ChIP analysis was performed in PKCα-KR cells in the absence (binding region 1 (black squares), 2 (inverted triangles), 3 (white squares)) and presence of 200 nM mithramycin (mtm; binding region 1 (triangles), 2 (diamonds), 3 (circles)) for 12 h to determine Sp1 binding at the *prkcb* promoter (average of *n* = 3 biological replicates); (**D**) The expression levels of Sp1 and PKCβII were determined by Western blotting in MIEV and PKCα-KR cells treated with 200 nM mtm (representative blot shown of *n* = 3 experiments; densitometry of the blots and the full blots shown in Appendix A). (**E**) qPCR analysis of *prkcb*, *sp1*, *bcl2*, *vegfa*, *blnk*, *lef1* genes in PKCα-KR cells upon treatment with mtm (average of n = 2/3 biological replicates, an average of technical duplicates). *Gapdh* was used as the reference gene and normalised to NDC. Unpaired student *t*-tests (**B**,**C**) or one-way ANOVA (**E**) were used to analyse the data (where biological triplicates were performed). * *p* < 0.05, ** *p* < 0.01, *** *p* < 0.001, **** *p* < 0.0001.

**Figure 3 cancers-14-06006-f003:**
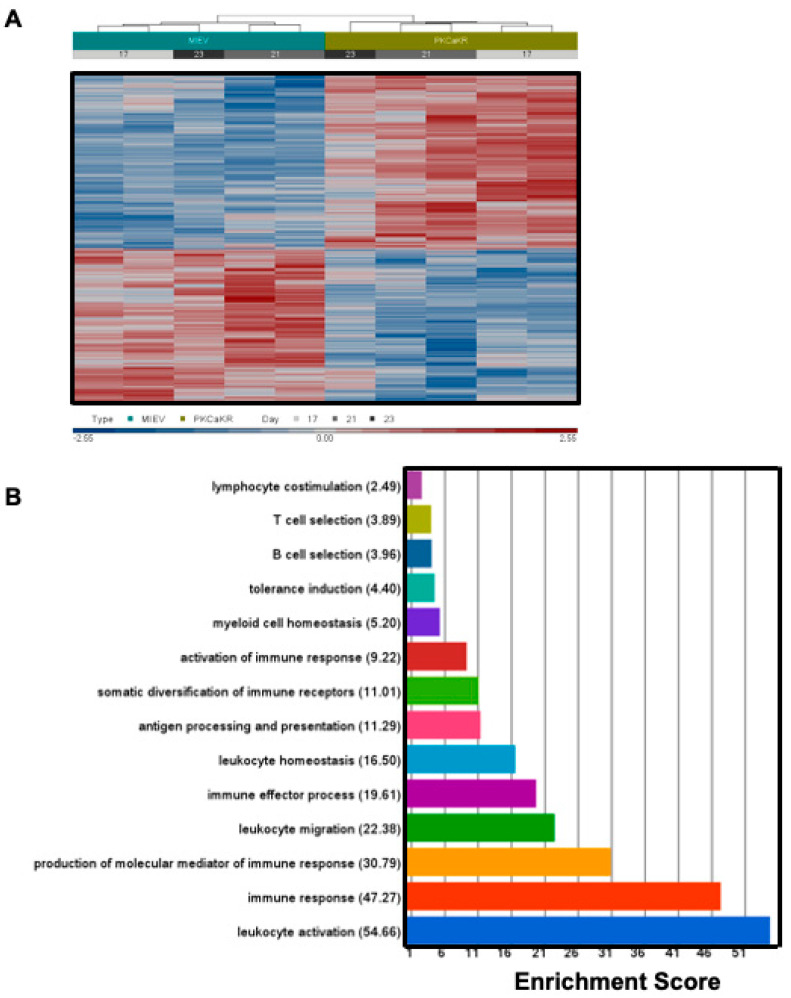
Gene expression comparisons between PKCα-KR vs. MIEV B lineage cells. (**A**) Hierarchical clustering of the relative gene expression using the Euclidian matric with average linkage (Partek Genomics Suite, v6.6) for PKCα-KR vs. MIEV raw Affymetrix files (*n* = 5). Culture day for each column identified in grey for both PKCα-KR and MIEV cells (day 17, 21 or 23). (**B**) Gene ontology enrichment analysis of the significantly altered genes (fold change ± 1.2 and *p*-value < 0.05) between late co-culture PKCα-KR vs. MIEV cells, identified immune system processes as the most enriched groups of pathways. Functional groups with the highest over-representation of genes within the gene list compared to background are shown alongside their respective enrichment score (−log *p*-value of a chi-square test).

**Figure 4 cancers-14-06006-f004:**
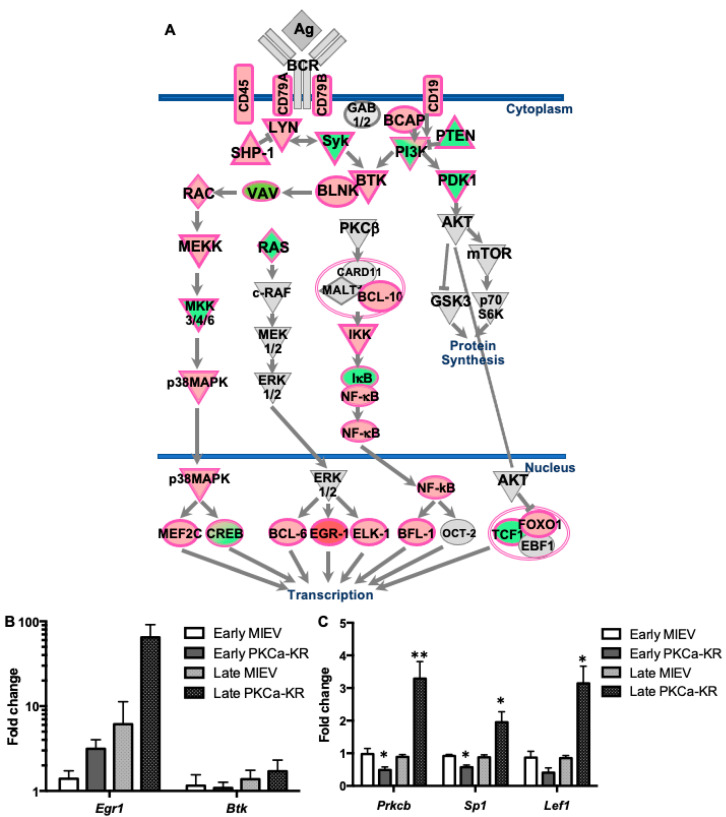
Global gene expression analysis revealed activation of the BCR-mediated signalling pathway in PKCα-KR cells. Global RNA analysis was performed using Affymetrix GeneChip mouse gene 1.0 ST on MIEV and PKCα-KR transduced cells at d17–23 in the B cell transformation co-culture. (**A**) Dysregulated components of the BCR pathway in PKCα-KR vs. MIEV cells isolated from late co-cultures (*n* = 5 PKCα-KR, *n* = 5 MIEV) were identified. Significantly up- and down-regulated components are highlighted in pink/red and green, respectively; (**B**) qPCR validation of *egr1* and *btk* genes, which were both upregulated in the microarray analysis at the late stage (d15–23) of B cell transformation, compared with the early stage (d6–10). Data are an average of *n* = 2–4 biological replicates, normalised to *tbp*. (**C**) Comparison of *prkcb*, *sp1* and *lef1* gene expression in the early vs. late stages of B cell transformation co-cultures, determined by qPCR. Data represent *n* = 3–5 biological replicates, normalised to *gapdh*. Unpaired student *t*-tests were used to analyse the data. * *p* < 0.05, ** *p* < 0.01.

**Figure 5 cancers-14-06006-f005:**
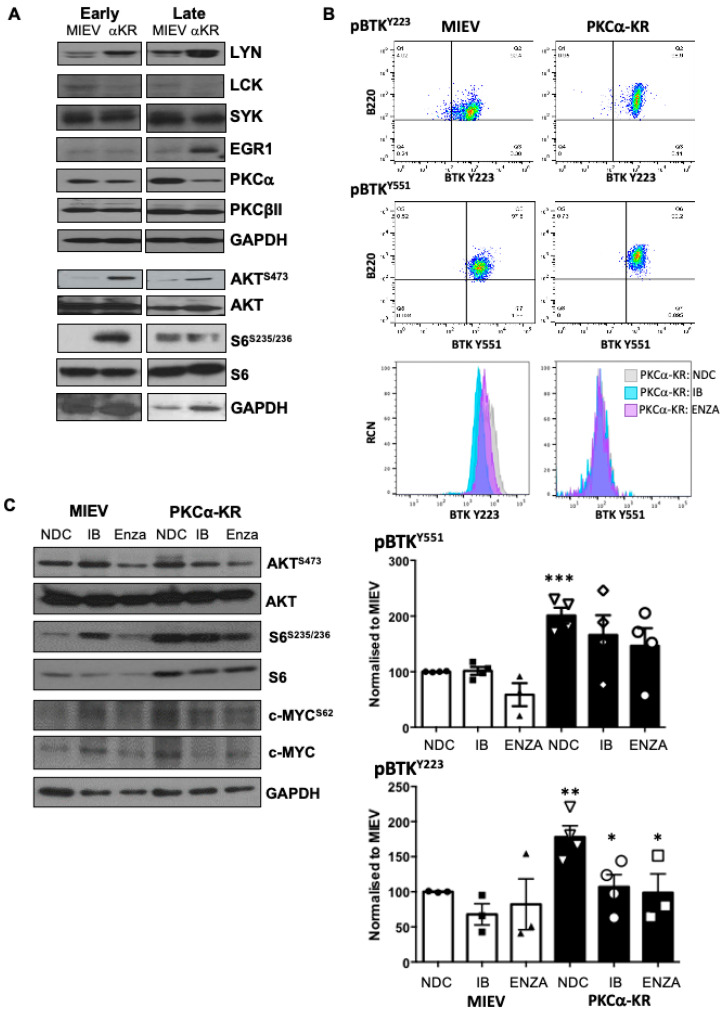
Inhibition of PKCβ activity downregulates key proteins in the BCR signalosome in mouse CLL model. (**A**) The expression/activation status of key hubs proximal and distal to the BCR pathway was analysed by Western blotting in early and late co-cultures of MIEV and PKCα-KR cells. Representative Western blots are shown. Densitometry of Western blots for these proteins (*n* ≥ 4) is shown in Appendix A. (**B**) Phospho-flow was used to analyse the levels of BTK^Y551^ and BTK^Y223^ phosphorylation in MIEV (left) and PKCα-KR (right) cells in the presence and absence of either Ibrutinib (IB; 1 μM) or enzastaurin (Enza; 20 μM). Upper panel shows the flow cytometry dot plots of the mouse B lineage cells (B220^+^) vs. phospho-BTK, gated on FSC/SSC, while lower histogram plots show the effect of drug treatments on BTK^Y551^ and BTK^Y223^ phosphorylation as overlay (NDC—pale grey; IB—turquoise; ENZA—pink). Lower graphs show the average MFI of phospho-BTK normalised to MIEV no drug control (NDC; *n* = 3/4 independent experiments; MIEV—white bars (NDC (circles), IB (squares), ENZA (triangles), PKCα-KR—black bars (NDC (inverted triangles), IB (circles), ENZA (squares)). (**C**) MIEV and PKCα-KR cells were treated with either IB or Enza, or NDC. Representative Western blots are shown identifying the effect on key proteins within the BCR signalosome as indicated. Densitometry of the Western blots for these proteins (*n* = 3) is shown in Appendix A. One-way ANOVA was used to analyse the data. * *p* < 0.05, ** *p* < 0.01, *** *p* < 0.001.

**Figure 6 cancers-14-06006-f006:**
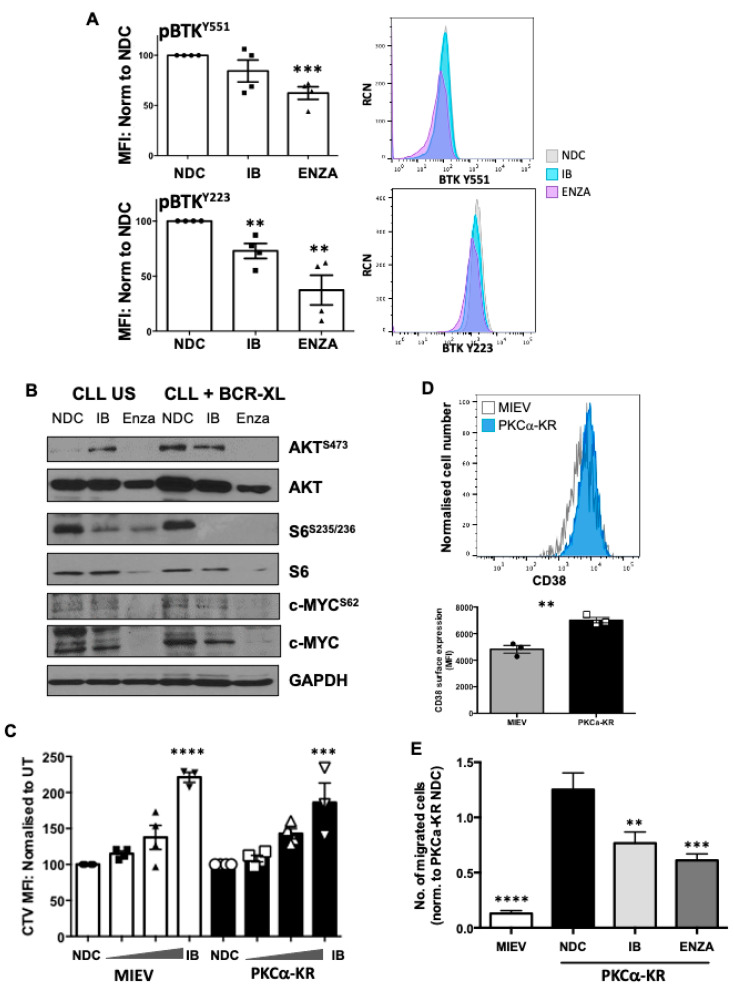
Inhibition of PKCβ upon treatment with enzastaurin downregulates key proteins within the BCR signalosome in primary CLL cells. (**A**,**B**) Human CLL cells were treated with 1 μM IB or 10 μM Enza in the presence or absence of BCR crosslinking (BCR-XL; F(ab’)_2_ fragment stimulation). A. Phospho-flow was used to analyse the levels of BTK^Y551^ and BTK^Y223^ phosphorylation upon treatment with drugs ± BCR-XL. Left-Graphs show the average MFI of the individual phospho-BTK sites as indicated, normalised to MIEV NDC (*n* = 4 individual patient samples; NDC (circles), IB (squares), ENZA (triangles)). Right-Histogram plots show the effect of drug treatments on BTK^Y551^ and BTK^Y223^ phosphorylation as overlay (NDC—pale grey; IB—turquoise; ENZA—pink); (**B**) Western blots were performed to identify the effect of drug treatment on key proteins within the BCR signalosome in human CLL cells. Representative Western blots are shown. Densitometry of the Western blots for these proteins (*n* ≥ 3 individual patients) is shown in Appendix A; (**C**) Late co-culture MIEV and PKCα-KR cells (2 × 10^6^) were labelled with CTV and cultured for 48 h in the presence (100 nM—3 μM) of increasing concentrations of IB (squares, triangles, inverted triangles) or NDC (circles). Results are expressed as the CTV MFI relative to NDC cells for MIEV and PKCα-KR cultures (*n* = 4 individual experiments); (**D**) A histogram is shown comparing the surface expression of CD38 in MIEV and PKCα-KR cells taken from d33 of co-culture (upper) and the average MFI of CD38 expression is shown relative to MIEV cultures (*n* = 3); (**E**) Migration assessment of MIEV and PKCα-KR cells was performed in the presence and absence of 1 μM IB or 20 μM Enza. The data shown represent an average of 5 independent experiments. Unpaired student *t*-tests (**D**) or one-way ANOVA (**A**,**C**,**E**) were used to analyse the data. ** *p* < 0.01, *** *p* < 0.001, **** *p* < 0.0001.

## Data Availability

Microarray data presented in Figure 3 are available at the Gene Expression Omnibus (GEO) repository (GSE185075). The microarray data presented in Appendix A are available at the GEO repository (GSE210348).

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
