# Peer review of "PKCβ Facilitates Leukemogenesis in Chronic Lymphocytic Leukaemia by Promoting Constitutive BCR-Mediated Signalling"

_cancers, 2022, doi:10.3390/cancers14236006_

Round 1

Reviewer 1 Report

Hay et al in their manuscript „PKCbeta facilitates leukemogenesis in chronic lymphocytic leukaemia by promoting constitutive BCR-mediated signaling “ set out to better define the roles of PKCalpha and PKCbeta genes, respectivly, in a model of CLL.   

Previous work in mouse cell and in vivo models showed that a reduction of PKCalpha functionality resulted in PKCbetaII overexpression and the development of an aggressive form of CLL, while PKCbeta knock-down in the Eµ-TCL1 mouse model led to an inhibition of leukemic cells. In addition, overexpression of PKCbetaII per se in B-cells was shown to not lead to malignant disease. Based on these findings the authors used a PKCalpha kinase inactive mouse cell model with and without lentiviral knock down of PKCbeta in order to study resulting effects on the B-cell receptor pathway.

They observed that, based on the PKCalpha inactive background, PKCbeta is important for the onset of leukemic development. They found in PKCalpha inactivated cells an upregulation of the prkcb gene which was due to SP1 binding to several promoter binding sites. This was accompanied by an upregulation of genes inivolved in B-cell activation and signaling, but also of genes involved in AKT/mTOR signaling. These findings were reflected in primary cells from CLL patients, which not only were impacted with regard to BCR activity but also migratory behavour.

Hay and colleagues systematically explored the impact of either PKCalpha inactivation alone, or in combination with PKCbeta knock-down in their cell model, use pathway specific inhibitors to differentiate between the contributions of these pathways to the observed effects, and extend their findings from mouse to human leukemic cells. The manuscript is kept short, their data are presented concisely and appropriately discussed, the relevant literature is being cited. 

All Supplementary Data (Tables and Figures) were missing from the provided files for reviewing this manuscript and I would expect them for reviewing a revision. 

I appreciate, however, the uncut pictures of the western blots highlighting the sections used for the Figures (present in duplicates)!

Minor points: 

I would like to know why and how the 2 different reference genes for the various qPCRs were selected, and why a multiple reference gene approach was not used. 

Also, I would ask the authors to outline the tests to determine efficiency of real-time primers. 

In addition, I found several abbreviations for real-time PCR (RT-PCR, qRT PCR, qPCR) – I suggest using just one term throughout the manuscript.

Author Response

1.1 All Supplementary Data (Tables and Figures) were missing from the provided files for reviewing this manuscript and I would expect them for reviewing a revision. 

  • I apologise that the supplemental data was missing from the original submission. The supplementary data file now accompanies the revision.

1.2 I appreciate, however, the uncut pictures of the western blots highlighting the sections used for the Figures (present in duplicates)!

  • At the editors request, we have now incorporated the uncut western blot images into the supplementary data file.

Minor points: 

1.3 I would like to know why and how the 2 different reference genes for the various qPCRs were selected, and why a multiple reference gene approach was not used. 

  • The choice of reference gene chosen is dependent on the Ct value of the reference gene and the stability of the Ct value with different drug/cell conditions. In this study, we used both Sybr Green qPCR in which we designed our own primer pairs (Figure 4B) and Taqman qPCR where we used inventoried primer sets (Figures 1E, 2E, 4C): Tbp and gapdh were used as reference genes respectively in this study. In our laboratory we have characterised the stability of a number of different reference genes including gusb, tbp, 18S, gapdh, actin, the use of which are dependent on the model. Some reference genes are interchangeable and some are more suited to particular cellular systems. Therefore, we did not use multiple reference genes in each experiment.

1.4 Also, I would ask the authors to outline the tests to determine efficiency of real-time primers. 

  • When designing primer pairs, consideration was given to ensuring low GC content and the gene product spanning an intron. Optimisation of the primers was performed ensuring the correct annealing temperature was used. Primer specificity was determined by performing melt curves to ensure there was a single product. A list of our designed primers is included in the supplementary data file (Table S3), and we have added a note in the Materials and Methods section regarding their optimisation (line 146-147).

1.5 In addition, I found several abbreviations for real-time PCR (RT-PCR, qRT PCR, qPCR) – I suggest using just one term throughout the manuscript.

-    We have now altered these references to make everything the same – qPCR.

Reviewer 2 Report

In this manuscript, Hey et al. described the interconnection of PKC-beta expression with BCR signaling in Leukemogenesis in their aggressive mouse B-CLL-like model (PKC-alpha-KR). The finding is potentially engaging because of their mouse model of CLL to use as potential tools for preclinical therapeutic option evaluation in CLL.

The author here is trying to interpret that higher expression of PKC- beta correlates with mouse HPSC towards CLL development. For that, they first Knockdown in vitro the mouse HSCs with pkcrb sh RNA and then showed that after transduced PKCalpha-KR construct, these cells lose CD5 and CD23 expression. Also, retarded the growth compared to PKCalpha-KR cells in culture. Can the author give any data regarding the PKC-beta expression in different time points of culture PKCalpha-KR and PKCalpha-KR transduced with pkcrb shRNA, as PKC-beta is well characterized for its effect in B cell development, specifically B1 cells?

Did the author ever check(loss of CD5 and CD23) whether these cells differentiated in Plasma cells? What is the status of surface IgM in these Pkcrb KD cells? Are there any differences with Pkcalpha-KR cells?

Also, it would be good if the author could knockdown pkcrb in already transformed B-CLL. And check the outcome. That will clarify whether this PKC isoform is required for leukemic transformation or leukemia progression.

The author should incorporate the Isotype control and gating strategy in  Figure 1A.

Is there any specific reason author could show to use mithramycin instead of sp1 si-RNA in the CHIP experiment?

It is better if the author can represent the (Figure 2) CHIP data with the DNA Gel pictures(representative) to specify the INPUT, IgG, and IMMUNE, besides the statistical representation.

Table1 can be moved in the supplementary file, whereas in the primary manuscript author can produce microarray data with clustering analysis(pi chart or ven diagram etc.).

Author Response

2.1 Can the author give any data regarding the PKC-beta expression in different time points of culture PKCalpha-KR and PKCalpha-KR transduced with pkcrb shRNA, as PKC-beta is well characterized for its effect in B cell development, specifically B1 cells?

  • We have previously published the timeline for PKCb expression in the co-cultures, and more specifically PKCbII, during the PKCa-KR co-culture (Ref. 12 in the current manuscript; Nakagawa et al doi: 3324/haematol.2014.112276). We found that PKCbII was upregulated in the later stages of the PKCa-KR CLL-Like cells:OP9 co-culture, and provided the foundation of this current study. This study has been mentioned in the current manuscript (lines 70-72; 266-268). Our studies presented here demonstrate that we achieve a 70-80% downregulation of PKCb in the shRNA-PKCa-KR system (Figure 1E). Within the TCL1 mouse model it is considered that B1 cells are the cell of origin of the leukaemia. This is not clear in our cells in vitro, although they do express CD5 both in vitro and in vivo as we have previously published (Ref 11 in the current manuscript; Nakagawa et al DOI: 10.1158/0008-5472.CAN-05-0841).

2.2 Did the author ever check(loss of CD5 and CD23) whether these cells differentiated in Plasma cells? What is the status of surface IgM in these Pkcrb KD cells? Are there any differences with Pkcalpha-KR cells?

            - We have not checked this. In previous studies we have consistently noted the maintenance of Pax5 expression in B cells for the duration of the OP9 co-culture (up to d30-35), which goes hand in hand with CD19 expression. As Pax5 is downregulated with plasma cell differentiation, we considered that the B cells generated in our system were unable to differentiate into plasma cells. CD19 expression was maintained on sh-prkcb-PKCa-KR cells at the same levels as sh-SCR-PKCa-KR cells, as indicated in Figure 1B.

- We previously published that surface IgM expression is reduced in PKCa-KR (CLL-like) compared with MIEV cells (Ref 11; Nakagawa et al DOI: 10.1158/0008-5472.CAN-05-0841). Analysis of IgM surface expression showed no significant change between SCR-PKCa-KR and sh-prkcb-PKCa-KR cells (13.04±1.572 vs 11.78±0.843; p=0.578; n=5 individual experiments).

2.3 Also, it would be good if the author could knockdown pkcrb in already transformed B-CLL. And check the outcome. That will clarify whether this PKC isoform is required for leukemic transformation or leukemia progression.

- When we started this project, we attempted to KD prkcb after transforming the cells with PKCa-KR. This approach was unsuccessful because PKCa-KR cells proliferate naturally, therefore any residual, contaminating cells that had not undergone full prkcb KD grew and rapidly dominated the co-culture, making this point very difficult to address in our mouse model in vitro.

- In primary CLL cells, we have previously published that SP1 KD inhibits PKCbII expression, (Ref 17 in the current manuscript; Al-Sanabra et al DOI:10.1038/srep43228). These cells did not die, which suggests that PKCbII is not required for leukaemic progression. However primary human CLL cells do not naturally proliferate when isolated from peripheral blood. Our studies presented here indicate that if these cells were placed in pro-proliferative conditions they would exhibit reduced proliferation and survival, thereby inhibiting leukaemia progression, as seen in the mouse model.

- In summary, while this is a very interesting question to address, the situation in each of the models is quite complicated, and it would be a significant undertaking to address.

2.4 The author should incorporate the Isotype control and gating strategy in Figure 1A.

            - We now include unstained controls in the histograms shown in a revised Figure 1A and have added the gating strategy as a supplementary figure (Supplementary Figure S1). We have amended the text accordingly (lines 218-220).

2.5 Is there any specific reason author could show to use mithramycin instead of sp1 si-RNA in the CHIP experiment?

            - The PKCa-KR cells are very fast growing and so siRNA would be diluted out rapidly, whereas mithramycin treatment enables a more global approach. I appreciate that this may introduce some non-specific effects, however for these reasons we looked at the expression of SP1 targets that were identified in human CLL cells and we found that these were similarly downregulated, indicating that mithramycin selectively inhibited SP1 activity. These targets were in Supplementary Table S6 of the Supplementary Data file, which I am sorry you did not receive in the first review.

2.6 It is better if the author can represent the (Figure 2) CHIP data with the DNA Gel pictures (representative) to specify the INPUT, IgG, and IMMUNE, besides the statistical representation.

            - For analysis of our ChIP DNA samples, we used qPCR as it is a quantitative method which allows for the determination of fold enrichment within the purified, precipitated samples. Therefore we do not have DNA gel images to show.

2.7 Table1 can be moved in the supplementary file, whereas in the primary manuscript author can produce microarray data with clustering analysis (pi chart or ven diagram etc.).

            - As part of the supplementary data file we included additional analysis of the microarray to graphically represent Table 1 identifying the genes associated with the pathways. We have now swapped the data around, and present a heatmap and bar chart highlighting the dysregulated pathways identified in the microarray in the main text of the paper (Figure 3), and Table 1 has been moved to the supplementary data file as requested by the reviewer (Supplementary Table S7).

Reviewer 3 Report

The manuscript entitled “PKCB facilitates leukemogenesis in chronic lymphocytic leukaemia by promoting constitutive BCR-mediated signaling” describes an interesting and well-designed study. The manuscript describes the importance of PKCB expression in disease development in a mouse model, analyzes the mechanisms that promote PKCB upregulation, and outlines the signaling pathways downstream of BTK/PKCB. Some minor points should be addressed:

-       In the Material and Methods sub-section 2.2, the analyzed markers should be included in the description. Additionally, there is no mention of cell death analysis in the Material and Methods section.

-       Table 1 is too small and hard to read. It should be improved.

-       “Deregulated” should be replaced by “dysregulated”.

-       The densitometry analysis of western blots experiments needs to add to all Figures.

-       The files “cancers-1984579-original-images” and “cancers-1984579-supplementary” have the same information (original western blot images).

Author Response

3.1 In the Material and Methods sub-section 2.2, the analyzed markers should be included in the description. Additionally, there is no mention of cell death analysis in the Material and Methods section.

            - The antibodies used, and therefore the analysed markers are included in the Supplementary Data file (Table S2). We have added key markers into sub-section 2.2 (lines 94-96). We have now added information about the apoptosis assay as part of sub-section 2.5 (lines 118-119).

3.2 Table 1 is too small and hard to read. It should be improved.

- In response to the comments from Reviewer 2, we have moved Table 1 into the Supplementary Data file (Supplementary Table S7). This change has allowed us to increase the text shown in the Table.

3.3 “Deregulated” should be replaced by “dysregulated”.

            - This has been changed throughout the manuscript.

3.4 The densitometry analysis of western blots experiments needs to add to all Figures.

- This data was included in the supplementary data file, I apologise that this file was not included as part of the first review, we have now attached this file.         

3.5 The files “cancers-1984579-original-images” and “cancers-1984579-supplementary” have the same information (original western blot images).

- I apologise for this, as mentioned above we have now included the supplementary file.

Round 2

Reviewer 2 Report

2.1 Can the author give any data regarding the PKC-beta expression in different time points of culture PKCalpha-KR and PKCalpha-KR transduced with pkcrb shRNA, as PKC-beta is well characterized for its effect in B cell development, specifically B1 cells?

  • We have previously published the timeline for PKCb expression in the co-cultures, and more specifically PKCbII, during the PKCa-KR co-culture (Ref. 12 in the current manuscript; Nakagawa et al doi: 3324/haematol.2014.112276). We found that PKCbII was upregulated in the later stages of the PKCa-KR CLL-Like cells:OP9 co-culture, and provided the foundation of this current study. This study has been mentioned in the current manuscript (lines 70-72; 266-268). Our studies presented here demonstrate that we achieve a 70-80% downregulation of PKCb in the shRNA-PKCa-KR system (Figure 1E). Within the TCL1 mouse model it is considered that B1 cells are the cell of origin of the leukaemia. This is not clear in our cells in vitro, although they do express CD5 both in vitro and in vivo as we have previously published (Ref 11 in the current manuscript; Nakagawa et al DOI: 10.1158/0008-5472.CAN-05-0841).

2.1.1 Thanks, author, for providing this information. Knockdown/out PKC alpha/beta in peritoneal cavity B1/B2 from your wild-type ICR mouse model would be an excellent experiment to address this query in the future.  

2.2 Did the author ever check(loss of CD5 and CD23) whether these cells differentiated in Plasma cells? What is the status of surface IgM in these Pkcrb KD cells? Are there any differences with Pkcalpha-KR cells?

           - We have not checked this. In previous studies we have consistently noted the maintenance of Pax5 expression in B cells for the duration of the OP9 co-culture (up to d30-35), which goes hand in hand with CD19 expression. As Pax5 is downregulated with plasma cell differentiation, we considered that the B cells generated in our system were unable to differentiate into plasma cells. CD19 expression was maintained on sh-prkcb-PKCa-KR cells at the same levels as sh-SCR-PKCa-KR cells, as indicated in Figure 1B.

- We previously published that surface IgM expression is reduced in PKCa-KR (CLL-like) compared with MIEV cells (Ref 11; Nakagawa et al DOI: 10.1158/0008-5472.CAN-05-0841). Analysis of IgM surface expression showed no significant change between SCR-PKCa-KR and sh-prkcb-PKCa-KR cells (13.04±1.572 vs 11.78±0.843; p=0.578; n=5 individual experiments).

2.2.1 PAX5 level is indeed related to differentiation to plasma cells, but some contradictions in this relation are also observed like Grace J. Liu, et.al. J Exp Med 2 November 2020; 217 (11): e20200147. doi: https://doi.org/10.1084/jem.20200147. So the author can also check CD138 expression in their mouse model to ensure that fact.    

sIgM level reduced compared to MIEV cells. But you claimed here that PKCa-KR with aggressive CLL-like features increased the BCR signaling component. How can you correlate these two observations in your mouse model?

2.3 Also, it would be good if the author could knockdown pkcrb in already transformed B-CLL. And check the outcome. That will clarify whether this PKC isoform is required for leukemic transformation or leukemia progression.

- When we started this project, we attempted to KD prkcb after transforming the cells with PKCa-KR. This approach was unsuccessful because PKCa-KR cells proliferate naturally, therefore any residual, contaminating cells that had not undergone full prkcb KD grew and rapidly dominated the co-culture, making this point very difficult to address in our mouse model in vitro.

- In primary CLL cells, we have previously published that SP1 KD inhibits PKCbII expression, (Ref 17 in the current manuscript; Al-Sanabra et al DOI:10.1038/srep43228). These cells did not die, which suggests that PKCbII is not required for leukaemic progression. However primary human CLL cells do not naturally proliferate when isolated from peripheral blood. Our studies presented here indicate that if these cells were placed in pro-proliferative conditions they would exhibit reduced proliferation and survival, thereby inhibiting leukaemia progression, as seen in the mouse model.

- In summary, while this is a very interesting question to address, the situation in each of the models is quite complicated, and it would be a significant undertaking to address

.

2.3.1Thanks the author for clarifying the limitations of this mouse model.

2.4 The author should incorporate the Isotype control and gating strategy in Figure 1A.

            - We now include unstained controls in the histograms shown in a revised Figure 1A and have added the gating strategy as a supplementary figure (Supplementary Figure S1). We have amended the text accordingly (lines 218-220).

2.4.1. Thanks to author for providing the details.

2.5 Is there any specific reason author could show to use mithramycin instead of sp1 si-RNA in the CHIP experiment?

            - The PKCa-KR cells are very fast growing and so siRNA would be diluted out rapidly, whereas mithramycin treatment enables a more global approach. I appreciate that this may introduce some non-specific effects, however for these reasons we looked at the expression of SP1 targets that were identified in human CLL cells and we found that these were similarly downregulated, indicating that mithramycin selectively inhibited SP1 activity. These targets were in Supplementary Table S6 of the Supplementary Data file, which I am sorry you did not receive in the first review.

2.5.1 Thanks 

2.6 It is better if the author can represent the (Figure 2) CHIP data with the DNA Gel pictures (representative) to specify the INPUT, IgG, and IMMUNE, besides the statistical representation.

           - For analysis of our ChIP DNA samples, we used qPCR as it is a quantitative method which allows for the determination of fold enrichment within the purified, precipitated samples. Therefore we do not have DNA gel images to show.

2.6.1 Thanks to the author for displaying this information.

2.7 Table1 can be moved in the supplementary file, whereas in the primary manuscript author can produce microarray data with clustering analysis (pi chart or ven diagram etc.).

            - As part of the supplementary data file we included additional analysis of the microarray to graphically represent Table 1 identifying the genes associated with the pathways. We have now swapped the data around, and present a heatmap and bar chart highlighting the dysregulated pathways identified in the microarray in the main text of the paper (Figure 3), and Table 1 has been moved to the supplementary data file as requested by the reviewer (Supplementary Table S7).

2.7.1. Thanks to the author for rearrangement of the figures.